# Fear of breast cancer among young Spanish women: Factor structure and psychometric properties of the Champion breast cancer fear scale

**Aldo Aguirre-Camacho**[1]*, **Beatriz Hidalgo**[1,2], **Gustavo González-Cuevas**[3]

**1** Department of Psychology, School of Biomedical Sciences, European University of Madrid, Madrid, Spain, **2** Department of Psychology, Autonomous University of Madrid, Madrid, Spain, **3** College of Pharmacy, Idaho State University, Meridian, Idaho, United States of America

☯ These authors contributed equally to this work.
* aldo.aguirre.c@gmail.com

**Data Availability Statement:** All relevant data are within the paper and its Supporting information files.

## Abstract

Heightened fear of breast cancer (FBC) has been linked to increased distress following breast cancer diagnosis and to avoidance of mammography screening. To our knowledge, however, no studies have examined the nature of FBC exclusively among young females, even though they are overrepresented in media stories of breast cancer. Given that no instruments are available to assess FBC in the Spanish language, we sought to 1) evaluate the psychometric properties and factor structure of the Champion Breast Cancer Fear Scale (CBCFS), and 2) offer preliminary data on the nature of FBC among young women. Participants (N = 442, mean age = 21.17, range 17–35) completed the translated CBCFS (CBCFS-es) and the Spanish version of the Short Health Anxiety Inventory. The CBCFS-es demonstrated good concurrent validity, internal consistency, and test-retest reliability. Confirmatory factor analysis showed adequate fit to a one-factor solution. The majority of participants reported considerably high levels of FBC, as 25.34% and 59.73% of them scored above the moderate- and high-FBC cut-offs, respectively. Moreover, FBC could not be explained by general concerns regarding health and illness, given that levels of health anxiety were low. Implications for health education, research, and clinical practice are discussed.

## Introduction

A substantial research base suggests that breast cancer has received more media attention over the last three decades than any other type of cancer, and perhaps any other health condition [1–3]. Even though this has helped raise awareness of breast cancer's early signs and symptoms, and thus contributed to lowering mortality rates [4], some authors have pointed out that increased media attention has also had unwanted side effects [3]. Many studies have found that breast cancer stories often contain information that highlights the negative consequences

**Funding:** The author(s) received no specific funding for this work.

**Competing interests:** The authors have declared that no competing interests exist.

of being diagnosed, the unpredictability related to living with breast cancer, and that also misrepresents breast cancer as affecting women of lower age than in reality [5, 6]. Accordingly, it has been suggested that the "age of breast cancer awareness" has resulted in disproportionate fear, to a point where breast cancer seems to be dreaded even more than other types of conditions of worse prognosis and higher prevalence [3, 7, 8].

Fear of breast cancer, as well as breast cancer-related worry and anxiety, has been found to both promote and interfere with cancer-preventive behaviors (e.g. information seeking, attendance to mammogram screening) across different studies [9]. However, such inconsistency has been partly attributed to several conceptual and methodological issues. First, the terms "fear", "worry", and "anxiety" have been often used interchangeably in this area of research [9], even though some studies have distinguished between the affective nature of fear and the cognitive nature of worry and anxiety [10, 11]. Second, some authors have suggested that the relationship between fear/worry/anxiety and breast cancer screening may be best represented by an inverted-U, given that both low and high levels of fear/worry/anxiety seem to deter women from engaging in cancer-preventive behaviors [12, 13]. And third, these three constructs have been measured following different approaches within this area of research, including structured questions, single items, *ad hoc* checklists assessing barriers to cancer screening, generic measures of emotions (e.g. Symptoms of Anxiety and Depression Scale, State-Trait Anxiety Inventory), and cancer-specific questionnaires (e.g. Cancer Attitude Inventory, Cancer Worry Scale Revised for Genetic Counseling) [9, 14].

The development of the Champion Breast Cancer Fear Scale (CBCFS) constituted an important step forward in the assessment of fear of breast cancer, especially considering the conceptual and methodological inconsistencies found in this area of research [9, 10]. The CBCFS is based on a conceptual definition of fear as a negatively toned emotion (e.g. "*feeling* upset", "*feeling* anxious") accompanied by heightened physiological responses (e.g. "feeling jittery", "feeling one's heart beat faster"); items that reflected cognitive responses, rather than emotional and physiological ones, were excluded during the initial stages of development of the scale (i.e. "I don't like to *think* about breast cancer" and "The more you *think* about breast cancer, the more likely you are to get it"). The CBCFS is specifically centered on the threat posed by breast cancer, and thus it offers a clearer assessment of breast cancer fear in comparison to less specific measures.

The CBCFS has been used in many studies and validated in several cultures. The original one-factor structure of this scale was replicated by Moshki et al. [15] in a sample of Iranian women; however, a two-factor structure was obtained by Secginli [16] in a sample of Turkish women. The CBCFS was also adapted to assess fear of colorectal cancer in a Chinese sample [17], where the one factor structure was also supported. Details about the psychometric properties and factor structure of the CBCFS and subsequent cultural adaptations can be found in Table 1. To our knowledge, the CBCFS is the only scale that has been specifically developed to assess fear of breast cancer, and no Spanish version is available so far.

## The present study

The objectives of this study were 1) to translate the CBCFS into Spanish and analyze its psychometric properties and factor structure in a sample of young women attending university, and 2) to offer preliminary data on the nature of fear of breast cancer among women of this age group.

Fear of breast cancer has been most often studied among women of at least 40 years of age, in the context of breast cancer screening [9]. To our knowledge, however, no study has examined fear of breast cancer exclusively among very young women, even though doing so is

**Table 1. Basic study details, psychometric properties, and factor structure of the Champion breast cancer fear scale and subsequent cultural adaptations.**

|  | Champion et al. [10] | Secginli et al. [16] | Leung et al. [17]* | Moshki et al. [15] |
|---|---|---|---|---|
| Publication Year | 2004 | 2012 | 2014 | 2017 |
| Country | USA | Turkey | China | Iran |
| Sample Size | 1390 | 224 | 250 (53 male) | 482 |
| Participants with Cancer | No | No | No | No |
| Mean age (SD) | 66.1 (10.4) | 46.97 (6.68) | 75.3 (7.6) | 47.35 (9.81) |
| Scale mean (SD) | 21.18 (8.52) | 26.36 (7.29) | 3.10 (1.04)** | 26.29 (7.95) |
| Internal consistency | Cronbach $\alpha$ = .91 | Cronbach $\alpha$ = .90 | Cronbach $\alpha$ = .95 | Cronbach $\alpha$ = .95 |
| Test-Retest | $r$ = .70, $p$ < .001 | $r$ = .60, $p$ < .01 | $r$ = .52, $p$ = .001 | $r$ = .85, $p$ < .001 |
| Structural validity analysis | PCA | PCA | CFA | CFA |
| Factors | 1 | 2 | 1 | 1 |
| Explained Variance | 57% | 53.79% | N/A | 74.15% |
| Factor loadings range | .47–.84 | .62–.83 | .69 to .93 | .64 to .80 |

Note:

*In this study the Champion Breast Cancer Fear Scale was adapted to asses fear of colorectal cancer;

**The scale mean was calculated using the item mean, rather than the sum of the different items; PCA = Principal Components Analysis; CFA = Confirmatory Factor Analysis.

necessary for several reasons. First, it is important to delineate the extent to which younger women feel threatened by breast cancer. On one hand, young women may be less likely to fear breast cancer because of their objective lower risk [18]; also, the threat of breast cancer may be lower among newer generations, given that over the last two decades several nations have witnessed a stabilization and decline in breast cancer mortality rates [19, 20]. On the other hand, however, breast cancer continues to be quite unique in terms of the media attention it receives, and this may continue to create disproportionate alarm even among young females [2, 21, 22]. In fact, different studies have found that women under the age of 50 and even 40 are often represented in the mass media as the typical breast cancer patient/survivor [23, 24], even though they constitute a minority of those diagnosed with breast cancer [25]. Second, it is important to identify the specific factors that may explain the potential heightened fear of breast cancer among this age group. Breast cancer can threaten young women's perception of femininity and sexuality like no other health condition [26], and can put on hold important life goals, such as starting a career and a family. These are factors that may be especially salient to young women and thus increase their fear of breast cancer, considering that these factors are often highlighted in breast cancer news and stories wherein younger women are overrepresented [3, 10]. Third, a better understanding of the factors leading to heightened fear of breast cancer early on in women's lives may help in planning for more effective interventions directed at increasing breast-cancer preventive behaviours.

## Methods

### Participants

Participants were eligible if they were between 16 and 35 years of age and had never received a cancer diagnosis.

### Procedure

This study was approved by the Ethics Committee of the Autonomous University of Madrid (approval number: CEI 66–1181). Participants were recruited through a research participation

system, managed by the Faculty of Psychology. Participants who agreed to take part in the study gave their written informed consent and were offered course credits in exchange for their participation. A minority of participants from the participant pool were just under 18 years of age, however, the Ethics Committee approved the lack of parent or guardian consent. The process of adaptation and validation of the CBCFS comprised two stages: the translation from English to Spanish, and a validation survey.

**Translation from English to Spanish.** The translation process followed the *Dual Panel Method*; this approach to translating psychometric instruments was proposed by Swaine-Verdier et al. [27] to address some of the problems that may arise when using the *Forward/Backward Translation Method*. The Forward/Backward Translation Method is often considered the gold standard, however, no empirical evidence has been provided in support of such view. Moreover, the results of a recent review article of different procedures used in the cross-cultural adaptation of psychometric instruments suggests that back-translation is not a necessary step in the translation process [28].

In accordance with the Dual Panel Method, the initial translation was carried out by a bilingual panel of three women and three men; the purpose of this panel was to reach an accessible and conceptually equivalent translation. The final wording chosen for most items was based on general consensus, however, different alternatives were provided for two items (i.e. feeling upset and feeling uneasy) with potentially different translations in Spanish. This initial translation was then reviewed by a lay panel of five monolingual cancer-free Spanish women of average educational level. The purpose of this lay panel was to decide on the final version of the scale by ensuring that the translation provided by the bilingual panel was easy to understand and sounded natural in Spanish.

**Validation survey.** A survey was conducted to test the psychometric properties and factor structure of the Spanish version of the CBCFS (hereinafter CBCFS-es). Participants completed a questionnaire package that included questions about basic demographic characteristics and perceived health status, as well as the Short Health Anxiety Inventory, and the CBCFS-es. A subset of participants (N = 78) also completed the CBCFS-es a second time following two weeks.

## Measures

**Champion Breast Cancer Fear Scale (CBCFS).** The CFBSC [10] is a self-report 8-item measure of emotional and physiological responses to fear of breast cancer. Participants answers are provided using a 5-point Likert scale ranging from 1 ("strongly disagree") to 5 ("strongly agree"). The total score is a sum score of all 8 items. There are no reverse-scored items. According to the original study, fear of breast cancer can be conceptualized as low (score of 8 to 15), moderate (score of 16 to 23) and high (score of 24 to 40). Higher scores indicate higher levels of fear of breast cancer.

**Short Health Anxiety Inventory (SHAI).** The SHAI [29, 30] is a self-report 18-item measure of health anxiety independent of physical health status. The two-factor Spanish version of this scale was used in this study, which evaluates *health-related worry* and *feared consequences of having an illness*. Participants' responses are provided using a four-option multiple choice format ranging from 0 to 3 (i.e. no symptoms, mild symptoms, severe symptoms, and very severe symptoms, respectively). A cut-off score of 27 has been used to identify individuals with hypochondriasis and other anxiety disorders [31, 32]. Higher scores indicate higher levels of health anxiety. The internal consistency (Cronbach's α) of the SHAI in the current sample was .81.

## Statistical analyses

Descriptive statistics and validity analyses were conducted using SPSS version 23. Reliability analysis and confirmatory factor analysis were conducted in R (version 3.6.1.) using the *psych* package [33] and the *lavaan* package [34] respectively. The factorial model was fitted using diagonally-weighted least squares estimation (DWLS), as recommended when analyzing ordinal data [35]. Model fit was evaluated using a combination of exact fit and approximate fit indexes. Adequate model fit was defined by the following criteria: a non-significant chi-squared statistic value, Tucker-Lewis Index (TLI) equal to or above .95, Comparative Fit Index (CFI) equal to or above .95, Root Mean Square Error of Approximation (RMSEA) below .06, and Square Root Mean Residual (SRMR) below .08 [36].

Preliminary data analyses found a negligible amount of missing data in some variables (no more than 1.30%). After confirming these data were missing completely at random, the expectation-maximization (EM) algorithm was used to impute the missing values before carrying out the statistical analyses.

# Results

## Participants

The sample comprised 442 female undergraduate psychology students from the Autonomous University of Madrid, Spain. Participants' age ranged from 17 to 35 years and had a mean of 21.17 years (SD = 3.38). The overall health status of the sample was good, as most participants rated their health as either good (59.50%) or excellent (30.80%), whereas a minority rated it as fair (7.90%) and poor (1.80%). The majority of participants reported still living with their parents (84.20%) and attending university fulltime (86.40%); 14.30% also reported having a job.

**Translation from English to Spanish.**   The translation produced by the bilingual panel was well received by the lay panel. The final wording for items 3 and 7 were chosen based on what sounded more natural according to day-to-day usage in Spain (Table 2).

## Descriptive statistics

Table 3 shows the descriptive statistics of the study variables. It is worth noting that only 14.93% of participants showed levels of fear of breast cancer categorized as low.

## Reliability of the scale scores

The CBCFS-es demonstrated excellent internal consistency (Cronbach's α = .92, omega total ω = .92). Each of the 8 items also showed good corrected item-total correlations; these ranged

**Table 2. Items from the original and Spanish versions of the Champion breast cancer fear scale.**

| | |
|---|---|
| 1. The thought of breast cancer scares me | 1. El solo pensar en el cáncer de mama me asusta |
| 2. When I think about breast cancer, I feel nervous | 2. Pensar en el cáncer de mama me pone nerviosa |
| 3. When I think about breast cancer, I get upset | 3. Pensar en el cáncer de mama me estresa |
| 4. When I think about breast cancer, I get depressed | 4. Pensar en el cáncer de mama me deprime |
| 5. When I think about breast cancer, I get jittery | 5. Pensar en el cáncer de mama hace que me sienta intranquila |
| 6. When I think about breast cancer, my heart beats faster | 6. Pensar en el cáncer de mama me acelera el corazón |
| 7. When I think about breast cancer, I feel uneasy | 7. Pensar en el cáncer de mama me perturba |
| 8. When I think about breast cancer, I feel anxious | 8. Pensar en el cáncer de mama me produce ansiedad |

**Table 3. Descriptive statistics of study variables (N = 442).**

|  | Mean (SD) | Range | Scale range |
|---|---|---|---|
| Mean SHAI score | 15.61 (5.81) | 3–41 | 0–54 |
| Mean *FI score* | 13.40 (5.33) | 2–38 | 0–42 |
| Mean *CI score* | 2.21 (1.52) | 0–8 | 0–12 |
| Mean CBCFS-es score | 24.92 (7.85) | 8–40 | 8–40 |
|  |  |  | N (%) |
| Low fear of breast cancer (score 8–15) |  |  | 66 (14.93) |
| Moderate fear of breast cancer (score 16–23) |  |  | 112 (25.34) |
| High fear of breast cancer (score 24–40) |  |  | 264 (59.73) |

*Note*. SHAI = Short Health Anxiety scale; FI = *fear of illness* subscale; CI = *consequences of illness* subscale; CBCFS-es = Spanish version of Champion Breast Cancer Fear Scale. Higher scores on the SHAI, FI, and CI indicate higher levels of health anxiety, fear of illness, and consequences of illness, respectively; Higher scores on the CBCFS-es indicate higher levels of fear of breast cancer. Low, moderate and high fear of breast cancer as categorized by the CBCFS-es cutoffs.

from .67 to 78, well above the criterion of .30 recommended by Nunnally and Bernstein [37] but also slightly above the criterion of .70 that identifies items as potentially redundant (Table 4). The scale also demonstrated good stability over a two-week interval as assessed in a subsample of 78 participants; the test-retest reliability coefficient was .84.

## Concurrent validity

The CBCFS-es showed weak to moderate but statistically significant correlations with the subscales *fear of illness* ($r = .36$, $p < .001$) and *consequences of illness*, ($r = .25$, $p < .001$) from the SHAI.

## Structural validity

A one-factor model was fitted to the data to assess the unidimensionality of the CBCFS-es. The model failed the exact fit test ($\chi2 = 56.79$, df = 20, $p < .001$); this result however was expected because the chi-square test is usually affected by large sample sizes [38]. Approximate fit indexes indicate an adequate fit of the one-factor model to the data: CFI = 0.99, TLI = 0.99,

**Table 4. Means, standard deviations, percentage of floor/ceiling effects, and corrected item-total correlations of each of the items from the Spanish version of the Champion breast cancer fear scale (N = 442).**

| Items | M (SD) | Floor % | Ceiling % | CIT-C |
|---|---|---|---|---|
| 1. The thought of breast cancer scares me | 3.63 (1.13) | 5.4 | 20.9 | .71 |
| 2. When I think about breast cancer, I feel nervous | 3.25 (1.18) | 8.4 | 13.6 | .77 |
| 3. When I think about breast cancer, I get upset | 3.03 (1.22) | 12.0 | 12.4 | .74 |
| 4. When I think about breast cancer, I get depressed | 3.09 (1.27) | 14.0 | 14.2 | .67 |
| 5. When I think about breast cancer, I get jittery | 3.03 (1.23) | 12.4 | 11.8 | .74 |
| 6. When I think about breast cancer, my heart beats faster | 2.85 (1.26) | 18.0 | 9.1 | .74 |
| 7. When I think about breast cancer, I feel uneasy | 2.88 (1.25) | 17.0 | 10.6 | .76 |
| 8. When I think about breast cancer, I feel anxious | 3.18 (1.29) | 12.9 | 16.3 | .78 |

*Note*: CIT-C = Corrected Item-to-Total Coefficient; Floor % = Percentage of participants scoring the lowest on a given item; Ceiling % = Percentage of participants scoring the highest on a given item; Items are rated on a 5-point Likert-type scale (from 1 to 5).

**Table 5. Standardized factor loading estimates, standardized error variances and R-squared values for the one-factor model (N = 442).**

| Item | Estimate | Error variance | R-squared |
|---|---|---|---|
| 1.The thought of breast cancer scares me | .74 (.03) | .45 (.09) | .55 |
| 2.When I think about breast cancer, I feel nervous | .81 (.03) | .34 (.09) | .65 |
| 3.When I think about breast cancer, I get upset | .77 (.03) | .40 (.09) | .59 |
| 4.When I think about breast cancer, I get depressed | .70 (.03) | .51 (.09) | .48 |
| 5.When I think about breast cancer, I get jittery | .77 (.03) | .41 (.09) | .59 |
| 6.When I think about breast cancer, my heart beats faster | .77 (.03) | .40 (.09) | .60 |
| 7.When I think about breast cancer, I feel uneasy | .80 (.03) | .36 (.10) | .64 |
| 8.When I think about breast cancer, I feel anxious | .81 (.03) | .35 (.10) | .65 |

*Note*: Standard error of estimate in parentheses.

RMSEA = 0.065 (90% CI = 0.045, 0.085), SRMR = 0.035. As shown in Table 5, the values for the standardized factor loadings range from .70 to .81, and the proportion of variance explained by the factor in each item ranges from .48 to .65. These results indicate a strong relationship between the latent factor and the items of the scale.

We attempted to estimate the two-factor structure found by Secginli [16], but this model could not be fitted to the data due to the high estimated correlation between the two factors ($r > 1.00$). This result indicates that the two factors are not statistically distinguishable in the collected data and that they should be combined into one factor, which further supports the unidimensionality of the scale.

After assessing the dimensionality of the scale, we calculated the factor scores based on the one-factor solution that showed a good fit to the data. The one-factor model yielded factor scores that were almost indistinguishable from the sum scores ($r = .99$), which supports the use of sum scores for this scale as proposed in the original study [10].

## Discussion

The first objective of the present study was to translate the CBCFS from English to Spanish and assess its psychometric properties and factor structure in a sample of young Spanish women. The results showed that the adaptation process was successful. All of the items worked well in the analyses, as indicated by their high item-total correlations with the overall score and strong relationship with the latent factor; the scale as a whole also worked well, as indicated by its high internal consistency and test-retest reliability. The results of the CFA were congruent with the one-factor structure obtained by Champion et al. [10] in their study of development of the CBCFS, as well as with other adaptation studies [15, 17].

The results obtained here are consistent with the conceptualization of fear of breast cancer on which the CBCFS is based. The CBCFS-es total score was weakly to moderately correlated with the *fear of illness* and *consequences of illness* subscales from the SHAI, which can be explained in at least two ways. First, the CBCFS only comprises items related to emotional and physiological responses, whereas the SHAI mostly assesses cognitive responses (e.g. ability to control thoughts, imagining being sick, wondering about the meaning of bodily sensations) [29, 30]. This finding is congruent with those of previous studies showing that, although somehow related, cancer-related affect and cognition constitute separate factors with distinct roles in the process of influencing cancer-related health behaviors [9, 11]. Second, the SHAI assess anxiety about illness in general, whereas the CBCFS specifically assesses fear of breast cancer; that is, fear of breast cancer may be partly explained by a general propensity to show concern

for one's health but it seems to be also explained by factors specifically related to having breast cancer.

This study also sought to preliminarily examine the extent to which young Spanish women felt afraid of breast cancer, and the results suggest that they did to a considerable extent. The level of fear of breast cancer reported by participants was similarly high to that shown by samples of older women [10, 15, 39]; also, almost 60% of scores were above the cut-off indicating high levels of fear of breast cancer. In contrast, the level of health anxiety was relatively low: the distribution of the SHAI scores was positively skewed and the sample's mean (i.e. 15.61) was well below the cut-off of 27, which may indicate the presence of hypochondriasis and other anxiety disorders. Altogether, these results suggest that participants were little concerned about becoming ill in general, and yet, they seemed considerably afraid of breast cancer. This finding suggest that breast cancer may be quite unique in terms of the amount fear it evokes, even among young women without significant health concerns and with low objective risk. This is in line with the results of previous studies [2, 3, 40]. In addition, when compared to previous studies carried out over the last two decades, these results also suggest that the level of fear evoked by breast cancer has not changed much, despite the positive epidemiological changes that have taken place during this time in several developed nations (i.e. reduced mortality and improved treatments) [41].

The results of this study should be interpreted in light of some limitations. The sample was composed of university students, and they may differ from young women not attending university in terms of their socio-economic status and that of their families. Socio-economic status (SES) has been linked to health literacy and health-related attitudes and behaviors [42], and therefore it may also be related to the reported levels of fear of breast cancer. Moreover, university students may differ from non-university students in their level of knowledge about breast cancer, given their higher educational level and likely that of their families. Nevertheless, both university and non-university students may be equally exposed to inaccurate and fatalistic media information regarding breast cancer, which may reduce any of such potential differences. In any case, the relevance of these factors (i.e. SES, health literacy, level of knowledge and media information about breast cancer) in relation to fear of breast cancer are, to our knowledge, yet to be studied among women of this age group; this also appears to be the first study examining fear of breast cancer within this cultural context. Therefore, the results presented here should be taken with caution, given the preliminary nature of these findings. Accordingly, future studies should examine this topic further; special attention should be devoted to uncovering the specific factors that may lead to heightened fear of breast cancer early on in women's lives.

We believe that the findings presented here have important implications for work in health education and prevention, as well as for research and clinical practice. First, it is important to identify the factors that may lead to high levels of fear of breast cancer among women of this age group. Doing so may help plan for more effective media campaigns or educational interventions wherein any information gaps can be properly addressed early on in women's lives, without creating unnecessary concern about breast cancer. Previous studies have in fact shown that the inaccurate/alarmist representation of breast cancer information can increase doubts among young women as to whether they should also be regularly screened, often putting in question recommendation for mammography screening [24]. In Spain breast cancer screening is based on recommendations from the European Union: all women between the ages of 50 and 69 are eligible for biannual mammograms, whereas women with a family history of breast cancer are also screened between the ages of 45 and 49 [43]. Nonetheless, fear of breast cancer has also been linked to other preventive behaviours aside from attendance to mammography

screening, and thus, it is also important to know whether young women may engage in such behaviours. For example, also based on the recommendations cited above, breast self-examination is recommended from ages 18 to 20, and a yearly clinical breast examination is recommended from the age of 25 onwards. Second, heightened fear of breast cancer may divert women's attention from other diseases that pose a similarly high or even higher risk to their health (e.g. heart disease). Previous studies have found that individuals who see themselves at risk for a specific disease tend to worry less about other health conditions [44, 45]. Therefore, young women may benefit from health education interventions that present them with a realistic view of the risk posed by breast cancer and other health conditions. Third, some of the misconceptions leading to heightened of fear of breast cancer early on in women's lives may also interfere with preventive behaviours. For example, the inaccurate/alarmist representation of breast cancer in the mass media can lead to fatalistic beliefs about breast cancer (e.g. that there is little one can do after receiving a breast cancer diagnosis) [46]. In turn, previous studies have found that individuals who hold fatalist beliefs about cancer are more likely to avoid information about cancer, which increases the probability that such fatalist beliefs remain unchallenged [47], and less likely to see the importance of engaging early detection practices [48].

In sum, the results of this study show evidence that the CBCFS-es is a valid and reliable instrument for assessment of fear of breast cancer among young Spanish women, even though this instrument had never been used with women as young as the ones in this sample. Nonetheless, with the exception of Secginli [16], the results obtained here do not differ from previous validation studies of the CBCFS carried out with samples of older women from different cultural contexts. Also, these results pose new questions regarding the nature of fear of breast cancer among young women and suggest this is an issue that should be explored further in future studies given the implications discussed above.

## Supporting information

**S1 File. Data file corresponding to the study.**
(SAV)

## Author Contributions

**Conceptualization:** Aldo Aguirre-Camacho, Beatriz Hidalgo.

**Data curation:** Aldo Aguirre-Camacho, Beatriz Hidalgo, Gustavo González-Cuevas.

**Formal analysis:** Aldo Aguirre-Camacho, Beatriz Hidalgo, Gustavo González-Cuevas.

**Investigation:** Aldo Aguirre-Camacho.

**Methodology:** Aldo Aguirre-Camacho, Beatriz Hidalgo, Gustavo González-Cuevas.

**Project administration:** Aldo Aguirre-Camacho.

**Resources:** Aldo Aguirre-Camacho.

**Supervision:** Aldo Aguirre-Camacho, Beatriz Hidalgo, Gustavo González-Cuevas.

**Writing – original draft:** Aldo Aguirre-Camacho.

**Writing – review & editing:** Aldo Aguirre-Camacho, Beatriz Hidalgo, Gustavo González-Cuevas.

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
