## [Decision Letter · Decision Letter 0]

12 Feb 2021

PONE-D-20-36556

Fear of breast cancer among young Spanish women: Factor structure and psychometric properties of the Champion’s fear of breast cancer scale

PLOS ONE

Dear Dr. Aguirre-Camacho,

Thank you for submitting your manuscript to PLOS ONE. After careful consideration, we feel that it has merit but does not fully meet PLOS ONE’s publication criteria as it currently stands. Therefore, we invite you to submit a revised version of the manuscript that addresses the points raised during the review process.

We look forward to receiving your revised manuscript.

Kind regards,

César Leal-Costa, Ph. D

Academic Editor

PLOS ONE

Journal Requirements:

2. We note you have included a table to which you do not refer in the text of your manuscript. Please ensure that you refer to Table 3 in your text; if accepted, production will need this reference to link the reader to the Table.

Reviewers' comments:

Reviewer's Responses to Questions

**Comments to the Author**

1. Is the manuscript technically sound, and do the data support the conclusions?

Reviewer #1: Partly

Reviewer #2: Yes

2. Has the statistical analysis been performed appropriately and rigorously? 

Reviewer #1: Yes

Reviewer #2: Yes

3. Have the authors made all data underlying the findings in their manuscript fully available?

Reviewer #1: Yes

Reviewer #2: Yes

4. Is the manuscript presented in an intelligible fashion and written in standard English?

Reviewer #1: Yes

Reviewer #2: Yes

5. Review Comments to the Author

Reviewer #1: In this manuscript the validation of a scale to assess ‘Fear of Breast cáncer among Young spanish women’ is presented. The manuscript is well written but there are some problems that I highlight below:

1. A very profuse introduction to the fear of breast cancer is presented in the first pages. However, this introduction does not provide data on psychometric studies that have been done on the same scale in other cultures, although these studies are cited on page 6, lines 121 to 123. There is also no information on other scales that can measure fear or anxiety even if they are not designed for this type of disease.

2. Apart from the title of the manuscript, nothing in the introduction indicates to the reader anticipate in the introduction that it is a psychometric work until page 6 where the objective is exposed. However a psychometric study should: 1) introduce the research problem, and 2) present all psychometric properties obtained in other cultures with the same scale, and similar scales that can provide information on the construct in this type of disease. In this sense, since one of the studies obtained a two-dimensional scale and the other studies are one-dimensional, the authors should draw attention to this discordant result and test both dimensional structures.

3. This study employs a forward and backward translation procedure so much sui generis. The fact that the initial team consisting of three women and three men does not guarantee that the translation of the instrument is correct. The correct procedure is that 1) a Spanish native (or more) accredited in English translate from English into Spanish, and 2) another English native (or more) with accredited knowledge of Spanish to translate from Spanish into English. So both versions in English (original and translated) are compared and the differences are resolved by the research team and the translators. The procedure used in this study does not guarantee a correct translation. Of course, the translation of item 1 is quite unfortunate if it was used with that wording.

4. Although it appears among the limitations of the study, a serious deficiency is that the collected sample does not contain clinical cases (i.e. women with breast cancer) and is also a university sample, which clearly signs any generalization of the results of the study.

5. Statment such as 'The total score is a sum score of all 8 items' must be justified. Of course, as long as the dimensionality of the scale is not known, it is not possible to make such a statement. In any case, the fact that a set of items load into a specific dimension does not justify adding the scores obtained on each item to obtain a total score. These scores are counts, not measures.

6. The original version has items with five categories. However, it would have been very interesting to research whether or not this number of categories is appropriate in the Spanish version. There are measurement models that can shed a lot of light on the appropriate number of categories on a scale, and whether the distance between categories is well established.

7. This study does not investigate the ceiling and floor effect on items and total scores. When there is ceiling and floor effect, the reliability of the scores is severely threatened.

8. On page 11, line 218 the following title appears 'Scale reliability'. It would be desirable to be careful with this statement, since in the classic test model, scales (psychometric tests) are not reliable or valid. Reliability and validity is from scores and can change (in fact they do) from sample to sample. Therefore, it is more appropriate to talk about reliability of scores.

9. You should be more careful when moving a result from the table to the text. For example, the table shows that 14.93% had low scores, but in the text (page 10, line 208) 14.89% appears.

10. All items have homogeneity indices above .30, but the recommendation of Nunnally and Berstein and many other references in psychometrics is that those values should not go beyond .70. A homogeneity index above that value is an indicator that each item alone serves to measure construct, and all other items only provide redundant information. That is, you have to be careful with the interpretations of these homogeneity indices.

11. Page 11, line 223. ‘the Pearson product-moment corelation coeficient’ can be described as ‘test-retest reliability coefficient’.

12. Page 12, line 232, the title is 'convergent validity'. The correct title should be 'concurrent validity'. A convergent validity coefficient can only arise from a multimethod-multitrait matrix, and that study is not presented in this manuscript.

13. Page 12, line 232 says 'factor loadings range from .74 to .86' when you should say '... range from .74 to .81'.

14. Page 13, line 243 say ‘… ranges from .48 to .73’ when the correct results are ‘ranges from .48 to .65’.

15. Lines 250 through 253 on page 13 attempt to explain why it is not possible to test a two-factor solution. In my opinion, the results should be offered for future readers to interpret themselves.

16. Finally, reference 46 does not appear in the reference list.

Reviewer #2: The theme of the article is innovative and brings really very interesting information to the topic of study, as well as clinical implications for the population under study. I encourage the authors to continue in this line.

To improve the quality of the article I make the following suggestions:

INTRODUCTION:

In this paper, authors are focused about fear of breast cancer among women. Is it possible to explain more about previous studies in this field (with Spanish samples or about another type of cancer)

Is this the first study about this topic? Can we find cultural differences about fear of breast cancer?

DISCUSSION: Extend about limitations such as sample (psychology students) and clinical implications.

6. PLOS authors have the option to publish the peer review history of their article (what does this mean?). If published, this will include your full peer review and any attached files.

Reviewer #1: No

Reviewer #2: **Yes: **María José Quiles Sebastián

---

## [Author Response · Author response to Decision Letter 0]

27 Feb 2021

Reviewer #1: 

In this manuscript the validation of a scale to assess ‘Fear of Breast cancer among Young Spanish women’ is presented. The manuscript is well written but there are some problems that I highlight below:

1. A very profuse introduction to the fear of breast cancer is presented in the first pages. However, this introduction does not provide data on psychometric studies that have been done on the same scale in other cultures, although these studies are cited on page 6, lines 121 to 123. There is also no information on other scales that can measure fear or anxiety even if they are not designed for this type of disease.

-We have reorganized and reduced the length of the introduction. The introduction now includes information regarding the different approaches that had been used to assess fear/worry/anxiety related to breast cancer before the development of the Champion Breast Cancer Fear Scale (CBCFS), which can be found at the end of the second paragraph (oage 3, line 63 onwards). Also, the introduction now includes data on the psychometric properties and factor structure of the CBCFS and subsequent validation studies; this information can be found in the fourth paragraph (page 4, line 80) and the newly added Table 1 (page 5).

2. Apart from the title of the manuscript, nothing in the introduction indicates to the reader anticipate in the introduction that it is a psychometric work until page 6 where the objective is exposed. However a psychometric study should: 1) introduce the research problem, and 2) present all psychometric properties obtained in other cultures with the same scale, and similar scales that can provide information on the construct in this type of disease. In this sense, since one of the studies obtained a two-dimensional scale and the other studies are one-dimensional, the authors should draw attention to this discordant result and test both dimensional structures.

-We have reorganized the introduction to highlight this is mainly a psychometric work. We introduce and describe the CBCFS in the third paragraph (page 4, line 69) and the subsequent cultural adaptations of this scale in the fourth paragraph (page 4, line 80). At the end of the fourth paragraph we hint what the research problem is, namely, that currently no Spanish version of the CBCFS is available. The objective of the study is now introduced right after this, in the fifth paragraph of the introduction (page 5, line 97). Again, information about the psychometric properties obtained in other cultures with the same scale can now be found in Table 1. We also tested both dimensional structures (one- and two-factors) obtained in previous studies and include information about these analyses at the very end of the Results section (page 14, lines 259-263).

3. This study employs a forward and backward translation procedure so much sui generis. The fact that the initial team consisting of three women and three men does not guarantee that the translation of the instrument is correct. The correct procedure is that 1) a Spanish native (or more) accredited in English translate from English into Spanish, and 2) another English native (or more) with accredited knowledge of Spanish to translate from Spanish into English. So both versions in English (original and translated) are compared and the differences are resolved by the research team and the translators. The procedure used in this study does not guarantee a correct translation. Of course, the translation of item 1 is quite unfortunate if it was used with that wording.

-The Forward/Backward Translation Method is often referred to as “the gold standard” in the translation of psychometric instruments. However, there is really no evidence in support of this view (Acquadro et al., 2008; Epstein et al., 1015) (please see references 27 and 28). Therefore, with all due respect, we do not agree with the reviewer that the Forward/Backward Translation Method would have constituted “the correct procedure” to follow.

The translation procedure used in this study (i.e. The Dual Panel Method) constitutes an entirely different approach and one of the existing alternatives to using the Forward/Backward Translation Method. Swaine‐Verdier et al. (2004) (please see reference 27) make a strong case against using the “Forward/Backward Translation Method”. Instead, they describe The Dual Panel Method, which has been extensively used in previous research. 

We have further clarified this in the text (page 7, lines 139 – 146).

Acquadro C, Conway K, Hareendran A, Aaronson N, European Regulatory Issues and Quality of Life Assessment (ERIQA) Group. Literature review of methods to translate health‐related quality of life questionnaires for use in multinational clinical trials. Value in Health. 2008 May;11(3):509-21.

Epstein, J., Santo, R. M., & Guillemin, F. (2015). A review of guidelines for cross-cultural adaptation of questionnaires could not bring out a consensus. Journal of clinical epidemiology, 68(4), 435-441.

Swaine-Verdier A, Doward LC, Hagell P, Thorsen H, McKenna SP. Adapting quality of life instruments. Value in health. 2004 Sep 1;7:S27-30.

4. Although it appears among the limitations of the study, a serious deficiency is that the collected sample does not contain clinical cases (i.e. women with breast cancer) and is also a university sample, which clearly signs any generalization of the results of the study.

-With all due respect, we do not agree with the reviewer that the absence of clinical cases in the sample constitutes a “serious deficiency”. As we describe in the sixth paragraph from the introduction (page 5, line 101), the vast majority of studies examining fear of breast cancer have been conducted with samples of cancer-free women in the context of breast cancer screening (please see reference 9). All previous versions of the CBCFS have also been validated within samples of cancer-free women in the context of breast cancer screening.

Nonetheless, we do agree with the reviewer in that a sample of university students constitutes a limitation, and we do acknowledge and discuss why this may be the case. However, arguably, the significance of this limitation also depends on the type of studies that use this scale in the future. For example, the sample used in this study would be very different compared to a sample of older women in the context of breast cancer screening, or a sample of women diagnosed with breast cancer. However, we do plan to continue using this scale within samples of women of the same age range as the one used in this study, as we are specifically interested in learning about the nature of fear of breast cancer early on in women’s life. Nevertheless, we understand that we would still need to conduct further analyses to ensure the scale works well.

5. Statments such as 'The total score is a sum score of all 8 items' must be justified. Of course, as long as the dimensionality of the scale is not known, it is not possible to make such a statement. In any case, the fact that a set of items load into a specific dimension does not justify adding the scores obtained on each item to obtain a total score. These scores are counts, not measures.

-We kept the scoring method that was used in the original study given that our intention was validating the original instrument. However, we agree with the point raised by the reviewer, using a sum score (or an alternative scoring model) should be adequately justified. We have calculated the factor scores (considering the one-factor model that was fitted to the data) and have obtained an almost perfect correlation between the two types of scores (r= 0.99) which supports the use of sum scoring for this scale. We have added this additional analysis to the manuscript (page 14, lines 264-267)

6. The original version has items with five categories. However, it would have been very interesting to research whether or not this number of categories is appropriate in the Spanish version. There are measurement models that can shed a lot of light on the appropriate number of categories on a scale, and whether the distance between categories is well established.

-We decided to maintain the number of response categories that was used in the original scale for this validation. We did not consider exploring the possibility of using a different number as there is evidence that shows that, across scale types and demographic groups: 1) using less than 5 categories (i.e. using 2 to 4 categories) results in poorer reliability, and 2) there is no clear psychometric advantage in using 6 or more response categories (Simms et al., 2019).

Simms, L. J., Zelazny, K., Williams, T. F., & Bernstein, L. (2019). Does the number of response options matter? Psychometric perspectives using personality questionnaire data. Psychological assessment, 31(4), 557.

7. This study does not investigate the ceiling and floor effect on items and total scores. When there is ceiling and floor effect, the reliability of the scores is severely threatened.

-In the initial stages of data analysis, we inspected the histograms and the skewness of the distribution of scores of each item to rule out the possibility of floor/ceiling effects that could impact the results. We agree with the reviewer that this information (the distribution of scores) is relevant and should be reported in the manuscript. We have now added additional information in Table 4 (page 13) regarding the percentage of people who scored in the lowest/highest value in each category. We did not find a significant proportion of responses at either the low or high end of the Likert scale.

8. On page 11, line 218 the following title appears 'Scale reliability'. It would be desirable to be careful with this statement, since in the classic test model, scales (psychometric tests) are not reliable or valid. Reliability and validity is from scores and can change (in fact they do) from sample to sample. Therefore, it is more appropriate to talk about reliability of scores.

-Indeed, reliability and validity are properties of the scores and not scale properties. We have changed this in the manuscript (page 13, line 241). 

9. You should be more careful when moving a result from the table to the text. For example, the table shows that 14.93% had low scores, but in the text (page 10, line 208) 14.89% appears.

-We apologize for this mistake. This has been corrected. Thank you for noticing!

10. All items have homogeneity indices above .30, but the recommendation of Nunnally and Berstein and many other references in psychometrics is that those values should not go beyond .70. A homogeneity index above that value is an indicator that each item alone serves to measure construct, and all other items only provide redundant information. That is, you have to be careful with the interpretations of these homogeneity indices.

-Yes, when these values reach levels beyond 0.7, we need to consider the possibility of items being redundant. We have added a clarification regarding the interpretation of these values in the manuscript (page 12, lines 227-230).

11. Page 11, line 223. ‘the Pearson product-moment corelation coeficient’ can be described as ‘test-retest reliability coefficient’.

Yes, the latter term is more appropriate for a scale validation. We have changed this in the manuscript (page 12, line 231).

12. Page 12, line 232, the title is 'convergent validity'. The correct title should be 'concurrent validity'. A convergent validity coefficient can only arise from a multimethod-multitrait matrix, and that study is not presented in this manuscript.

We have made this change in the manuscript (page 13, line 241).

13. Page 12, line 232 says 'factor loadings range from .74 to .86' when you should say '... range from .74 to .81'.

-We apologize for this mistake. According to Table 4, the factor loadings actually range from .70 to .81. This has been corrected in the text (page 12, line 242). Thank you for noticing!

14. Page 13, line 243 say ‘… ranges from .48 to .73’ when the correct results are ‘ranges from .48 to .65’.

-Again, we apologize for this mistake. This has been corrected. Thank you for noticing!

15. Lines 250 through 253 on page 13 attempt to explain why it is not possible to test a two-factor solution. In my opinion, the results should be offered for future readers to interpret themselves.

-We added more information regarding the estimation of the two-factor model with our data. When attempting to estimate a two-factor structure with our data, the estimated correlations between factor 1 and factor 2 are too high (r > 1.00) which makes a two-factor model estimation infeasible. This is an indication that both factors are indistinguishable and should be combined into one single factor. We added this clarification to the manuscript (page 14, line 259). 

16. Finally, reference 46 does not appear in the reference list.

-We apologize for this mistake. There was not in fact a reference 46, given that we had skipped the citation 44 in the text. Thus, “[45,46]” has been changed to “[44,45]” in page 16 line 326. Thank you for noticing! 

Reviewer #2: 

The theme of the article is innovative and brings really very interesting information to the topic of study, as well as clinical implications for the population under study. I encourage the authors to continue in this line.

We thank the reviewer for her comments and encouragement!

To improve the quality of the article I make the following suggestions:

INTRODUCTION:

1-In this paper, authors are focused about fear of breast cancer among women. Is it possible to explain more about previous studies in this field (with Spanish samples or about another type of cancer)

To our knowledge, no previous studies had focused on fear of breast cancer (or any other type of cancer) specifically among young women. As we mention in the introduction, fear of breast cancer has been most often studied among women of at least 40 years of age, in the context of breast cancer screening. In the second paragraph of the introduction (page 3, line 53), we mention the relevance of fear of breast cancer for breast cancer screening, as well as the conceptual and methodological problems found in previous studies within this area of research. We also clarify that, to our knowledge, fear of breast cancer had not been studied within Spanish samples. For these reasons, we do present the reasons as to why it is important to study fear of breast cancer among young women (page 5, line 101).

2-Is this the first study about this topic? Can we find cultural differences about fear of breast cancer?

-Yes, to our knowledge this is the first study examining fear of breast cancer among very young women. Therefore, any potential cultural differences in fear of breast cancer among women of this age group are yet to be explored.

To our knowledge, fear of breast cancer among older women has not been studied from a cross-cultural perspective. Therefore, it is still not clear which cultural factors may account for heightened fear of breast cancer. Interest in examining fear of breast cancer has been mostly limited to determining its role as a factor interfering with or promoting screening behavior. 

DISCUSSION: 

3-Extend about limitations such as sample (psychology students) and clinical implications.

-We have extended both the paragraph on limitations (page 16, line 306) and clinical implications (page 17, line 323).

---

## [Decision Letter · Decision Letter 1]

22 Mar 2021

Fear of breast cancer among young Spanish women: factor structure and psychometric properties of the Champion breast cancer fear scale

PONE-D-20-36556R1

Dear Dr. Aguirre-Camacho,

We’re pleased to inform you that your manuscript has been judged scientifically suitable for publication and will be formally accepted for publication once it meets all outstanding technical requirements.

Kind regards,

César Leal-Costa, Ph. D

Academic Editor

PLOS ONE

Additional Editor Comments (optional):

Reviewers' comments:

Reviewer's Responses to Questions

**Comments to the Author**

1. If the authors have adequately addressed your comments raised in a previous round of review and you feel that this manuscript is now acceptable for publication, you may indicate that here to bypass the “Comments to the Author” section, enter your conflict of interest statement in the “Confidential to Editor” section, and submit your "Accept" recommendation.

Reviewer #1: All comments have been addressed

Reviewer #2: All comments have been addressed

2. Is the manuscript technically sound, and do the data support the conclusions?

Reviewer #1: Yes

Reviewer #2: Yes

3. Has the statistical analysis been performed appropriately and rigorously? 

Reviewer #1: Yes

Reviewer #2: Yes

4. Have the authors made all data underlying the findings in their manuscript fully available?

Reviewer #1: Yes

Reviewer #2: Yes

5. Is the manuscript presented in an intelligible fashion and written in standard English?

Reviewer #1: Yes

Reviewer #2: Yes

6. Review Comments to the Author

Reviewer #1: The authors have adequately responded to all the requirements made in the review. The procedure used for translating the scale into Spanish may be very novel but it is not clear that it is a clear overcoming of the forward-backward translation method. Psychometric analysis is sound within the traditional methodology used with factor analysis and the classic test model. The authors should have been bolder and use Rasch's model to investigate the number of appropriate categories on this scale. I would have given the manuscript a differential value.

Reviewer #2: Los autores han respondido de manera satisfactoria a las recomendaciones realizadas, por lo que considero que el manuscrito cumple satisfactoriamente los requerimientos de la revista y puede ser aceptado para su publicación.

7. PLOS authors have the option to publish the peer review history of their article (what does this mean?). If published, this will include your full peer review and any attached files.

Reviewer #1: No

Reviewer #2: **Yes: **María-José Quiles-Sebastián

---

## [Editor Report · Acceptance letter]

24 Mar 2021

PONE-D-20-36556R1 

Fear of breast cancer among young Spanish women: factor structure and psychometric properties of the Champion breast cancer fear scale 

Dear Dr. Aguirre-Camacho:

I'm pleased to inform you that your manuscript has been deemed suitable for publication in PLOS ONE. Congratulations! Your manuscript is now with our production department. 

Kind regards, 

on behalf of

Dr. César Leal-Costa 

Academic Editor

PLOS ONE